# Constitutive G protein coupling profiles of understudied orphan GPCRs

**Sumin Lu**[1], **Wonjo Jang**[1], **Asuka Inoue**[2], **Nevin A. Lambert**[1]*

**1** Department of Pharmacology and Toxicology, Medical College of Georgia, Augusta University, Augusta, Georgia, United States of America, **2** Graduate School of Pharmaceutical Sciences, Tohoku University, Sendai, Japan

* nelambert@augusta.edu

**Data Availability Statement:** All relevant data are within the manuscript and its Supporting Information files.

**Funding:** This work was supported by a grant from the National Institutes of Health (GM130142 to N.

## Abstract

A large number of GPCRs are potentially valuable drug targets but remain understudied. Many of these lack well-validated activating ligands and are considered "orphan" receptors, and G protein coupling profiles have not been defined for many orphan GPCRs. Here we asked if constitutive receptor activity can be used to determine G protein coupling profiles of orphan GPCRs. We monitored nucleotide-sensitive interactions between 48 understudied orphan GPCRs and five G proteins (240 combinations) using bioluminescence resonance energy transfer (BRET). No receptor ligands were used, but GDP was used as a common G protein ligand to disrupt receptor-G protein complexes. Constitutive BRET between the same receptors and β-arrestins was also measured. We found sufficient GDP-sensitive BRET to generate G protein coupling profiles for 22 of the 48 receptors we studied. Altogether we identified 48 coupled receptor-G protein pairs, many of which have not been described previously. We conclude that receptor-G protein complexes that form spontaneously in the absence of guanine nucleotides can be used to profile G protein coupling of constitutively-active GPCRs. This approach may prove useful for studying G protein coupling of other GPCRs for which activating ligands are not available.

## Introduction

G protein-coupled receptors (GPCRs) are the targets of a large fraction of clinically-useful drugs, and efforts to develop new drugs targeting GPCRs are ongoing [1]. Defining characteristics of GPCRs are the natural ligands that bind and activate each receptor, and the intracellular transducers (G proteins and arrestins) that propagate signals to downstream effectors [2]. Individual GPCRs can couple to several different G proteins from more than one G protein family. Because each of the four G protein families ($G_{s/olf}$, $G_{i/o}$, $G_{q/11}$, and $G_{12/13}$) activates different downstream effectors, GPCR-G protein coupling profiles have traditionally been determined using second messenger assays, most commonly those that measure intracellular cyclic AMP (cAMP) and calcium. Although these assays are robust and quite sensitive, crosstalk between pathways can complicate interpretation, and comparable second messenger assays are not available for all four families. G protein coupling can also be determined by more direct methods, such as [35S]GTPγS binding *in vitro*, but these methods are more difficult to

A.L.). A.I. was funded by the PRIME
JP19gm5910013, the LEAP JP19gm0010004 and
the BIND JP19am0101095 from the Japan Agency
for Medical Research and Development (AMED).
The funders had no role in study design, data
collection and analysis, decision to publish, or
preparation of the manuscript.

implement, particularly at scale across multiple G protein subtypes [3]. More recently, genetic, spectroscopic and luminometric assays have been developed that allow more direct assessment of G protein coupling profiles in living cells [4–8]. These assays can detect coupling to all four G protein families, avoid ambiguity due to signal crosstalk, and are efficient enough to allow profiling of a large number of GPCRs in parallel.

Several recent studies have used these methods to profile G protein coupling of GPCRs in response to activating ligands [9–12]. However, for a large number of so-called orphan GPCRs the natural ligand is either not known or not well-validated, and surrogate activating ligands are not available [13]. Therefore, studies profiling G protein coupling have generally not included orphan GPCRs. In a recent study we found that many GPCRs would spontaneously form complexes with cognate G proteins in the absence of guanine nucleotides, and these complexes were disrupted by the addition of GDP [14]. This is consistent with the known ability of constitutively-active GPCRs to activate G proteins in the absence of an agonist [15]. It occurred to us that the nucleotide-sensitivity of spontaneous GPCR-G protein complexes could be used to define coupling profiles of orphan GPCRs without using activating ligands. Here we test this idea using 48 orphan GPCRs, most of which have not been extensively studied or characterized. We find that approximately half of the receptors we studied possess sufficient constitutive activity to define a G protein coupling profile. These results may facilitate efforts aimed at understanding the physiological roles these receptors, and at discovering and validating new drugs acting at GPCRs.

## Materials and methods

### Materials

Trypsin, culture media, PBS, DPBS, penicillin/streptomycin and L-glutamine were from GIBCO (ThermoFisher Scientific, Waltham, MA, USA). PEI MAX was purchased from Polysciences Inc. (Warrington, PA, USA). Digitonin, apyrase and GDP were purchased from MilliporeSigma (St. Louis, MO, USA). Coelenterazine h was purchased from Nanolight Technologies (Pinetop, AZ, USA).

### Plasmid DNA constructs

GPCR coding sequences were provided by Bryan Roth (University of North Carolina, Chapel Hill, NC; PRESTO-Tango Kit—#1000000068, Addgene, Watertown), MA, USA) [16], except for GPR139, which was a gift from Kirill Martemyanov [17]. For each receptor the coding sequence was amplified with a common forward primer (corresponding to a cleavable signal sequence) and custom reverse primer (corresponding to the receptor C terminus) and ligated into a pRluc8-N1 cloning vector. All plasmid constructs were verified by Sanger sequencing. Plasmids encoding Venus-Kras, Venus-PTP1b, Venus-1-155-G$\gamma_1$, and Venus-155-239-G$\beta_1$ have been described previously [4,18]. G$\alpha$ subunit plasmids were purchased from cdna.org (Bloomsburg University, Bloomsburg, PA). Plasmids encoding Venus-$\beta$-arrestin-1 and -2 were a gift from Vsevolod Gurevich (Vanderbilt University, Nashville, TN, USA), and plasmids encoding the S1 subunit of pertussis toxin (PTX-S1) was kindly provided by Stephen R. Ikeda (NIAAA, Rockville, MD, USA).

### Cell culture and transfection

HEK 293 cells (CLS Cat# 300192/p777_HEK293, RRID:CVCL_0045; ATCC, Manassas, VA, USA) were propagated in plastic flasks and on 6-well plates according to the supplier's protocol. HEK 293 cells with targeted deletion of GNAS, *GNAL*, *GNAQ*, *GNA11*, *GNA12* and *GNA13* (G protein three family knockouts; 3GKO) were derived, authenticated and

propagated as previously described [19]. Cells were transfected in growth medium using linear polyethyleneimine MAX (PEI MAX; MW 40,000) at an nitrogen/phosphate ratio of 20 and were used for experiments 48 hours later. Up to 3.0 μg of plasmid DNA was transfected in each well of a 6-well plate. For G protein experiments 3GKO cells were transfected with a GPCR-Rluc8, Gα subunit, Venus-1-155-Gγ$_2$, Venus-155-239-Gβ$_1$, and pcDNA3.1(+) or PTX-S1 in a (1:10:5:5:5) ratio for a total of 2.6 μg of plasmid DNA in each well of a 6-well plate. For arrestin experiments HEK 293 cells were transfected with a GPCR-Rluc8, Venus-β-arrestin-1 or -2, GRK2 and GRK6 in a 1:10:5:5 ratio for a total of 2.1 μg of plasmid DNA. For trafficking experiments HEK 293 cells were transfected with a GPCR-Rluc8 and either Venus-Kras or Venus-PTP1b in a 1:10 ratio for a total of 1.1 μg of plasmid DNA.

## BRET assays

For G protein coupling experiments cells were washed twice with permeabilization buffer (KPS) containing 140 mM KCl, 10 mM NaCl, 1 mM MgCl$_2$, 0.1 mM KEGTA, 20 mM NaHEPES (pH 7.2), harvested by trituration, permeabilized in KPS buffer containing 10 μg ml$^{-1}$ high purity digitonin, and transferred to opaque black 96-well plates. Measurements were made from permeabilized cells supplemented either with 100 μM GDP or 2U ml$^{-1}$ apyrase. For arrestin and trafficking experiments cells were washed twice in PBS and harvested by trituration in DPBS. For all experiments 5 μM coelenterazine h was used as a substrate. For the experiments shown in Fig 1, permeabilized cells were supplemented with apyrase, and GDP (100 μM) was injected during continuous recording using a Polarstar Optima plate reader (BMG Labtech, Offenburg, Germany). All other measurements were made using a Mithras LB940 photon-counting plate reader (Berthold Technologies GmbH, Bad Wildbad, Germany). Raw BRET signals were calculated as the emission intensity at 520–545 nm divided by the emission intensity at 475–495 nm. Net BRET signals were calculated as the raw BRET signal minus the raw BRET signal measured from cells expressing only the Rluc8 donor.

## Statistical analysis

The data shown in Fig 1 represent the mean ± SD of 16 technical replicates from one exemplary experiment. Because background basal BRET differed for each G protein, in this experiment raw BRET values for each trace are normalized the average of the first ten data points of all of the traces for a particular G protein. The data shown in Fig 2 represent the average of three independent experiments, each performed in duplicate. G protein heat maps (Fig 3) represent the difference in the raw BRET ratios measured from cells incubated in presence and absence of GDP ($\Delta BRET_{GDP}$). Arrestin heat maps represent the basal net BRET. No hypothesis testing was performed and no claims of statistical significance are made. The threshold for assigning G protein coupling was determined by assuming that the majority of receptor-G protein pairs would be uncoupled, and that the $\Delta BRET_{GDP}$ values for these pairs would be randomly distributed around zero. Coupled pairs were detected as outliers from this distribution using the ROUT method [20] implemented in GraphPad Prism 8 (GraphPad Software, La Jolla, CA) with Q (the maximum false discovery rate) set to 1%, meaning fewer than 1% of the detected (coupled) pairs are expected to be false-positives. The same procedure was used to detect receptor-arrestin pairs.

## Results

### Receptor-G protein interactions

We have previously monitored direct interactions between GPCRs and G proteins using bioluminescence resonance energy transfer (BRET) between receptors fused to *Renilla* luciferase

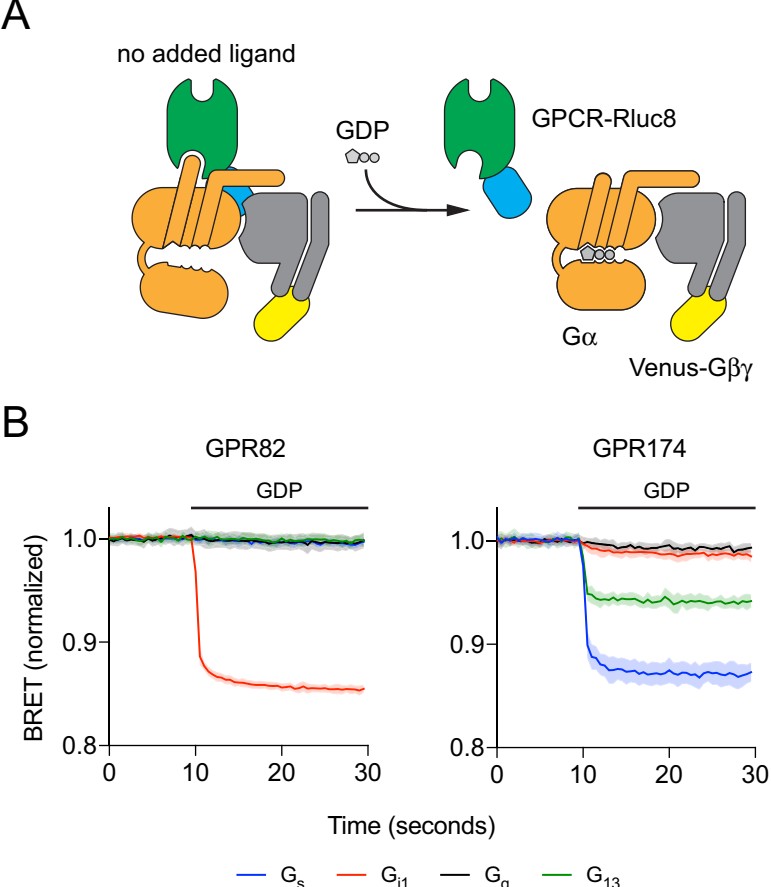

**Fig 1. Addition of GDP disrupts GPCR-G protein complexes.** (A) Cartoon representation of the experimental design. Constitutively-active GPCRs fused to *Renilla* luciferase (Rluc8) form spontaneous active-state complexes with nucleotide-free G protein heterotrimers fused (via the Gβγ subunit) to the fluorescent protein Venus in the absence of activating ligands. Addition of GDP (100 μM) disrupts these complexes, decreasing BRET between GPCR-Rluc8 and Gαβγ-Venus. (B) Representative experiments of this type with GPR82 and GPR174. Traces represent the mean ± SD of 16 technical replicates from a single experiment, and each trace is normalized to the basal BRET observed for that particular G protein. GDP was injected where indicated by the horizontal bar.

(Rluc8) and G protein heterotrimers tagged with the fluorescent protein Venus. Using permeabilized cells we found that many GPCRs spontaneously interacted with G proteins in a nucleotide-sensitive fashion [14]. Importantly, these constitutive GDP-sensitive interactions corresponded well to known G protein coupling, suggesting that it should be possible to study G protein coupling of orphan GPCRs without using activating ligands. To test this idea we fused Rluc8 to the C terminus of 48 class A orphan receptors, 43 of which are on the most recent list of understudied GPCR targets compiled by the Illuminating the Druggable Genome (IDG) project [21,22]. Receptors were coexpressed together with a Gα subunit and Venus-Gβγ in genome-edited HEK 293 cells lacking endogenous $G_{s/olf}$, $G_{q/11}$ and $G_{12/13}$ proteins [19]. We chose one Gα subunit to represent each of the four G protein families ($Gα_{i1}$, $Gα_s$-long, $Gα_q$, $Gα_{13}$) as well as $Gα_{15}$, due to its unique coupling properties [23]. Except when $Gα_{i1}$ was used, we also transfected the S1 subunit of pertussis toxin to prevent coupling of endogenous $G_{i/o}$ proteins to GPCRs.

Complexes between constitutively-active receptors and cognate G proteins formed spontaneously in permeabilized cells when apyrase was used to hydrolyze residual guanine

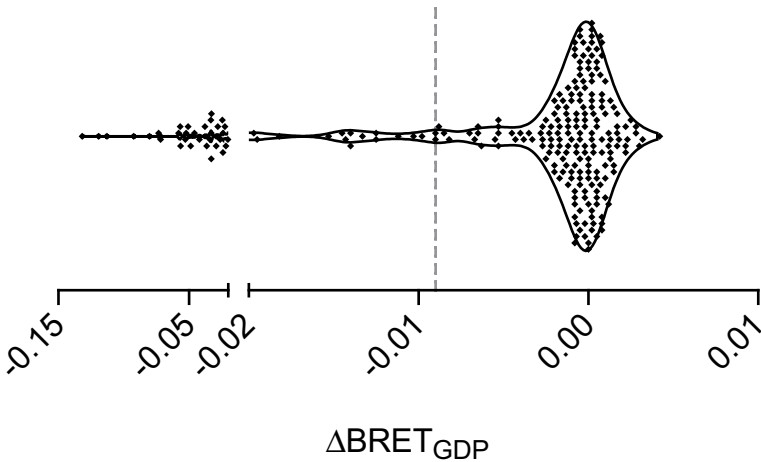

**Fig 2. Determination of threshold ΔBRET$_{GDP}$.** Most receptor-G protein pairs did not interact, and values of ΔBRET$_{GDP}$ were distributed around zero. Values to the left of the dashed vertical line (ΔBRET$_{GDP}$ = -0.009) were identified as outliers from this background distribution (i.e. coupled pairs) with a false discovery rate (FDR) of 1%. Each point represents a single receptor-G protein pair, and the mean of three independent experiments performed in duplicate.

nucleotides, thus maintaining the nucleotide-empty state of the G protein. Addition of GDP (100 μM) led to rapid complex dissociation, and a decrease in BRET between receptors and G proteins (Fig 1A). For example, GPR82 formed GDP-sensitive complexes with G$_{i1}$ heterotrimers, but not with G$_s$, G$_q$, or G$_{13}$ heterotrimers (Fig 1B). In contrast, GPR174 formed GDP-sensitive complexes primarily with G$_s$ and G$_{13}$ heterotrimers (Fig 1B). Almost nothing is known about GPR82 (Jensen PubMed Score 1.16), which is listed as a "probable" GPCR, and we were unable to find any reports of GPR82 coupling to G proteins in the literature. In contrast, GPR174 (Jensen PubMed Score 10.06) has been described as a receptor for lysophosphatidyl-L-serine (lysoPS) [7] and the chemokine CCL21 [24], and is known to couple to G$_{s/olf}$ and G$_{12/13}$ heterotrimers [9]. These results demonstrate the utility of this approach for profiling G protein coupling of orphan GPCRs.

Changes in BRET after addition of GDP (ΔBRET$_{GDP}$) for the 240 pairings in our sample clustered around zero, as expected if the majority of receptor-G protein pairs do not constitutively couple (Fig 2). However, a population of more negative values of ΔBRET$_{GDP}$ was apparent that presumably corresponds to coupled receptor-G protein pairs. We set a conservative threshold for coupling by identifying outliers from a random distribution of ΔBRET$_{GDP}$ values, using a false discovery rate (FDR) of 1% (see Materials and Methods).

Using this threshold G protein coupling was detected for 22 of the 48 receptors and 48 of the 240 pairings in our sample (Fig 3; S1 File). We detected constitutive coupling of 18 receptors to G$_{i1}$, 8 receptors to G$_s$, 6 receptors to G$_q$, 7 receptors to G$_{13}$, and 9 receptors to G$_{15}$. Of the 22 profiled receptors 11 are annotated for G protein coupling in the IUPHAR Guide to Pharmacology (GtoPdb), and within this set there was excellent agreement between our results and annotated coupling [13] (S1 File). The sole exception was GPR75, which coupled to G$_{i1}$ in our dataset but is annotated as coupling to G$_q$. This receptor has been shown to stimulate inositol phosphate turnover and calcium release in other studies [25]. In several cases, our results agreed with annotated receptor-G protein pairs, but also indicated coupling to additional G protein families. For example, our results confirmed coupling of GPR26 to G$_s$ [26], but indicated additional coupling to G$_{i1}$, G$_{15}$ and G$_q$. We also found several instances where no G protein coupling was annotated in GtoPdb, but where published reports indicated signaling

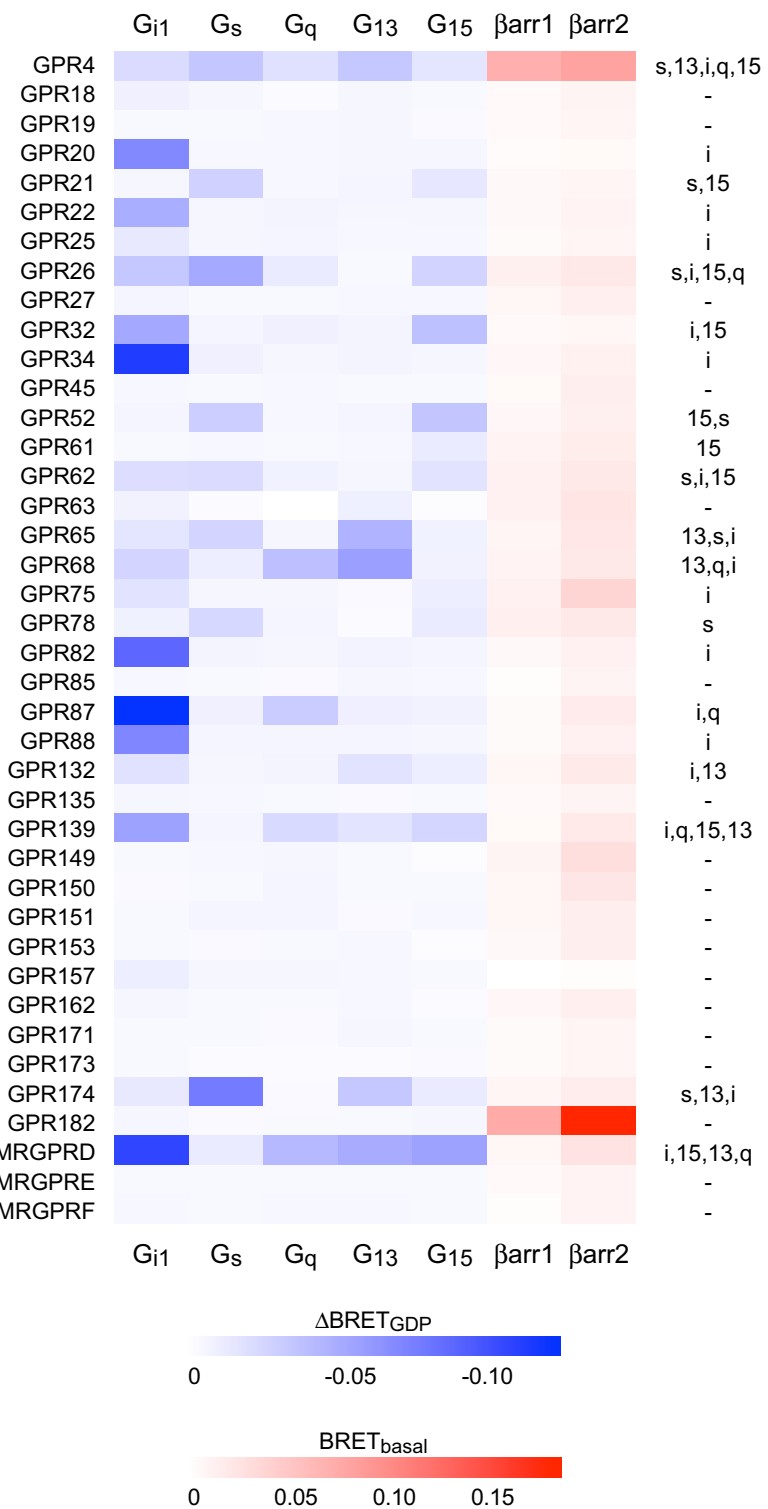

**Fig 3. Constitutive G protein and β-arrestin coupling of understudied GPCRs.** Heat maps representing the mean $\Delta BRET_{GDP}$ for 200 receptor-G protein pairs (blue) and basal net BRET for 80 receptor-β-arrestin pairs (red). The righthand column indicates the G proteins for which $\Delta BRET_{GDP}$ exceeded the determined threshold. Each cell represents the mean of three independent experiments performed in duplicate. Eight receptors that trafficked poorly to the plasma membrane are not shown here.

through a particular G protein pathway. In such cases our results were also generally in good agreement with previous reports, but again indicated coupling to additional G proteins that was previously unreported. For example, GPR62 has been shown to constitutively activate adenylyl cyclase (AC), albeit weakly, suggesting coupling to $G_s$ [27]. Our results confirm that this receptor couples to $G_s$, but also show similar coupling to $G_{i1}$ and $G_{15}$. Dual coupling to $G_i$ and $G_s$ proteins may help to explain relatively weak constitutive activation of AC by this receptor. These results illustrate the value of an unbiased profiling approach that includes G proteins from all four Gα subunit families.

For 26 of the receptors we studied $\Delta BRET_{GDP}$ did not meet threshold for any of the G proteins tested. The most likely explanation for this outcome is that these receptors simply lacked sufficient constitutive activity to couple efficiently to G proteins in the absence of a ligand. However, one alternative explanation is the failure of these receptors to traffic efficiently to the plasma membrane, where the majority of G protein heterotrimers are located. To test this idea we measured bystander BRET between each receptor and markers of the plasma membrane (PM) and endoplasmic reticulum (ER) [18]. Most receptors showed substantial BRET to the PM marker, and less BRET to the ER marker, indicating efficient trafficking to the cell surface. However, 8 receptors (GPR31, GPR37L1, GPR142, GPR146, GPR148, GPR152, GPR160 and MRGPRG) showed BRET to the ER marker that exceeded BRET to the PM marker, indicating inefficient trafficking to the PM (S1 File). All 8 of these receptors were among the 26 that failed to show constitutive G protein coupling, suggesting that retention of these receptors in the biosynthetic pathway may have contributed to our inability to detect G protein coupling.

## Receptor-arrestin interactions

It is also possible that some of the receptors that we studied do not couple to G proteins at all, as is the case for some "decoy" receptors (e.g. the C5a2 complement receptor) [28]. Because some decoy receptors bind to β-arrestins we asked if any of the orphan receptors in our sample interacted constitutively with these transducers by measuring basal BRET between receptors and Venus-β-arrestin-1 and Venus-β-arrestin-2 in intact cells. Basal BRET between unstimulated GPCRs and arrestins is typically low unless there is a specific interaction [29], or unless arrestins are recruited in some other way to membrane compartments where receptors are located. Accordingly, basal BRET between orphan receptors and Venus-β-arrestins was low for most of the receptors in our sample (Fig 3; S1 File). However, GPR182 and GPR4 were both outliers for both β-arrestin-1 and β-arrestin-2. GPR182 failed to couple detectably to G proteins, suggesting that this receptor may be biased towards interacting with arrestins rather than G proteins.

## Discussion

In the present study we measured guanine nucleotide-sensitive coupling of G proteins to a sample of understudied orphan GPCRs. We used an unbiased approach that directly indicates receptor association with unmodified Gα subunits and does not require an activating ligand. We were able to detect G protein coupling to 22 of the 48 receptors we studied. We confirmed many receptor-G protein pairings determined previously by other methods, and demonstrated several new pairings. With respect to the overall prevalence of coupling to different G protein subtypes, our results with constitutive activity of orphan receptors agree well with previous studies of agonist-induced coupling of non-orphan GPCRs [9–12]. $G_{i1}$ was the most frequent coupler (18 receptors), whereas the $G_{q/11}$ family (including $G_q$ and $G_{15}$) was the second-most frequent (15 receptors). Of the 9 receptors that coupled to only one G protein, 7 coupled solely to $G_{i1}$. We also found that coupling to $G_{13}$ (7 receptors) was more common in our dataset than

might be predicted based on GtoPdb annotations of all GPCRs, as shown previously by others [9,10]. It is possible that coupling to $G_{12/13}$ is underrepresented in GtoPdb because simple second messenger assays are not available for this family. Coupling to $G_{13}$ was always observed in conjunction with coupling to another G protein [30]. Among the $G_{13}$-coupled receptors in our sample were all 4 members of a closely-related family of acid-sensing receptors (GPR4, GPR65, GPR68 and GPR132) [31], all of which coupled to $G_{13}$ at least as well as any other G protein.

The assay that we used here has a particular advantage for studies of constitutive receptor activity, in that GDP can essentially be used as a common ligand to disrupt coupled GPCR-G protein complexes. This comes with a significant drawback, in that constitutive activity is required, and a subjective threshold was needed to assign receptor-G protein coupling. It is likely that many of the receptors that we were unable to profile will couple well to G proteins when bound to an activating ligand. Although these caveats mean that our study undoubtedly missed several receptor-G protein pairings, it also suggests that our results can help predict which of these orphan receptors have high and low constitutive activity. For example, GPR18 is a relatively well-studied receptor (Jensen PubMed Score 42.64) that binds to endogenous cannabinoid compounds [32] and is annotated in GtoPdb as coupling to $G_{i/o}$ and $G_{q/11}$. This receptor showed subthreshold $\Delta BRET_{GDP}$ (which was greatest for $G_{i1}$) in our study, suggesting that GPR18 has low constitutive activity compared to other receptors in our sample. Another limitation of our study is that we did not address selectivity among G proteins within a family, although this could be easily rectified with additional studies. We also identified several orphan receptors that are at least partly retained in the endoplasmic reticulum of HEK 293 cells. These receptors may require cell type-specific trafficking factors to reach the plasma membrane. For example, GPR37L1 is expressed almost exclusively in glial cells and is thought to couple to $G_{i/o}$ proteins [33], but trafficked poorly to the cell surface in HEK 293 cells.

We found one receptor, GPR182, that did not couple to G proteins in our assay, but did constitutively interact with β-arrestins. This result is consistent with a previous study that showed very high constitutive binding of a GPR182-V2R vasopressin receptor fusion protein to β-arrestin [16]. Gene-transcription studies suggest that this receptor may also signal through several canonical G protein pathways [34], but specific G protein coupling has not been reported. Given the demonstrated importance of GPR182 for cellular proliferation and hematopoiesis [35,36], our results suggest that further studies of GPR182 signaling mechanisms are warranted.

In summary, we were able to profile constitutive G protein coupling for a significant fraction of understudied class A orphan GPCRs. The success of this strategy suggests that it may be useful for profiling G protein coupling of other GPCRs (e.g. adhesion receptors and class C orphans) for which well-validated activating ligands are not available.

## Supporting information

**S1 File. Supplemental data file.**
(XLSX)

## Acknowledgments

We thank Steve Ikeda, Seva Gurevich, Kirill Martemyanov and Bryan Roth for providing plasmid DNA used in this study.

## Author Contributions

**Conceptualization:** Nevin A. Lambert.

**Data curation:** Sumin Lu, Wonjo Jang.

**Funding acquisition:** Asuka Inoue, Nevin A. Lambert.

**Investigation:** Sumin Lu, Wonjo Jang.

**Resources:** Asuka Inoue.

**Supervision:** Nevin A. Lambert.

**Writing – original draft:** Nevin A. Lambert.

**Writing – review & editing:** Wonjo Jang, Asuka Inoue, Nevin A. Lambert.

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
