## [Decision Letter · Decision Letter 0]

26 Mar 2021

Constitutive G protein coupling profiles of understudied orphan GPCRs

PONE-D-21-05345

Dear Dr. Lambert,

We’re pleased to inform you that your manuscript has been judged scientifically suitable for publication and will be formally accepted for publication once it meets all outstanding technical requirements.

Kind regards,

Erik C. Johnson

Academic Editor

PLOS ONE

Additional Editor Comments (optional):

Both reviewers were overall enthusiastic and positive about the manuscript, and I completely concur with them.  I congratulate the authors on a well-written manuscript on a thorough and well-performed study.   

Reviewers' comments:

Reviewer's Responses to Questions

**Comments to the Author**

1. Is the manuscript technically sound, and do the data support the conclusions?

Reviewer #1: Yes

Reviewer #2: Yes

2. Has the statistical analysis been performed appropriately and rigorously? 

Reviewer #1: Yes

Reviewer #2: Yes

3. Have the authors made all data underlying the findings in their manuscript fully available?

Reviewer #1: Yes

Reviewer #2: Yes

4. Is the manuscript presented in an intelligible fashion and written in standard English?

Reviewer #1: Yes

Reviewer #2: Yes

5. Review Comments to the Author

Reviewer #1: The Lambert group recently reported that some GPCRs spontaneously form complexes with cognate G proteins in the absence of guanine nucleotides, but these complexes could be disrupted by the addition of GDP. In this study the same group takes advantage of this phenomenon to profile G protein coupling of 48 orphan GPCRs (oGPCRs), for which agonist ligands are not known, by measuring nucleotide-sensitivity of spontaneous GPCR-G protein complexes by BRET. The authors find that approximately half (22 of 48) of the receptors studied possess some constitutive activity to define a G protein coupling profile to a member of either Gs, Gi, Gq and/or G12/13 family. Control experiments, such as bystander BRET, suggest most receptors were expressed at the plasma membrane, while some receptors that did not show a G protein-coupling profile were not expressed at the cell surface. Limitations of the proposed studies are that the BRET experiments required transfecting in 4 discrete DNA (receptor, Galpha, Gbeta and Ggamma subunits) and performed in cells that have been permeabilized. However, this concern of physiological relevance is mitigated by the fact that many of the receptor-G protein pairings were determined previously by other methods. What is also important here is that several new pairings are reported. However, whether these pairings indeed correspond to constitutively active GPCRs needs further follow-up in appropriate model systems. Also, there still remains 26 oGPCRs for which coupling was not observed, although these oGPCRs likely will couple to a G protein, but require binding to their cognate ligand to do so. In addition, the authors examined the oGPCRs for coupling to arrestin2 and arrestin3 by BRET, but only weak associations were found for all but two receptors (GPR4 & GPR182). This was somewhat predicted because ligand occupation is also a prerequisite for beta-arrestin recruitment to a GRK-phosphorylated GPCR. I found the supplemental data, which is an excel file with multiple sheets for all of the BRET data, well organized and useful. Overall this is an interesting and well executed study that will be of interest to the large GPCR community. There are no further concerns.

Reviewer #2: In this short report Lu and collaborators explored the basal coupling between 48 GPCRs and 5 G proteins. The authors proposed this procedure to identify the G protein selectivity of orphan GPCRs. The authors completed the study using BRET signal between ER and PM markers to identify if the GPCRs that failed to couple to G proteins trafficked correctly and also explore the basal coupling to all the GPCRs to betha arrestin. The paper is well written and the data even when is compelled in 3 figures is abundant and clearly presented. The data analysis and the methods are also clear. The discussion adds interesting information and critical thinking about the results in context.

6. PLOS authors have the option to publish the peer review history of their article (what does this mean?). If published, this will include your full peer review and any attached files.

Reviewer #1: No

Reviewer #2: No

---

## [Editor Report · Acceptance letter]

6 Apr 2021

PONE-D-21-05345 

Constitutive G protein coupling profiles of understudied orphan GPCRs 

Dear Dr. Lambert:

I'm pleased to inform you that your manuscript has been deemed suitable for publication in PLOS ONE. Congratulations! Your manuscript is now with our production department. 

Kind regards, 

on behalf of

Dr. Erik C. Johnson 

Academic Editor

PLOS ONE